# SMAC Mimetics Synergistically Cooperate with HDAC Inhibitors Enhancing TNF-α Autocrine Signaling

**DOI:** 10.3390/cancers15041315

**Published:** 2023-02-18

**Authors:** Yusuke Shibuya, Kei Kudo, Kristen P. Zeligs, David Anderson, Lidia Hernandez, Franklin Ning, Christopher B. Cole, Maria Fergusson, Noemi Kedei, John Lyons, Jason Taylor, Soumya Korrapati, Christina M. Annunziata

**Affiliations:** 1Women’s Malignancies Branch, National Cancer Institute, National Institutes of Health, Bethesda, MD 20892, USA; 2Department of Obstetrics and Gynecology, Division of Gynecology Oncology, Tohoku University School of Medicine, Miyagi 980-8574, Japan; 3Gynecologic Cancer Center of Excellence, Department of Obstetrics and Gynecology, Uniformed Services University and Walter Reed National Military Medical Center, Bethesda, MD 20814, USA; 4Department of Obstetrics, Gynecology and Reproductive Science, Division of Gynecologic Oncology, Icahn School of Medicine at Mount Sinai, New York, NY 10029, USA; 5Collaborative Protein Technology Resource, MD 20814, USA; 6Astex Pharmaceuticals, Cambridge, UK; 7Astex Pharmaceuticals, Pleasanton, CA 94588, USA

**Keywords:** ovarian cancer, SMAC mimetics, IAP inhibitor, HDAC inhibitor, TNF-α

## Abstract

**Simple Summary:**

The combination of a SMAC mimetic with an HDAC inhibitor is a novel and promising strategy for cancer treatment. The HDAC inhibitor mechanistically synergizes with SMAC mimetics by stimulating autocrine TNF-α production.

**Abstract:**

The overexpression of inhibitor of apoptosis (IAP) proteins is strongly related to poor survival of women with ovarian cancer. Recurrent ovarian cancers resist apoptosis due to the dysregulation of IAP proteins. Mechanistically, Second Mitochondrial Activator of Caspases (SMAC) mimetics suppress the functions of IAP proteins to restore apoptotic pathways resulting in tumor death. We previously conducted a phase 2 clinical trial of the single-agent SMAC mimetic birinapant and observed minimal drug response in women with recurrent ovarian cancer despite demonstrating on-target activity. Accordingly, we performed a high-throughput screening matrix to identify synergistic drug combinations with birinapant. SMAC mimetics in combination with an HDAC inhibitor showed remarkable synergy and was, therefore, selected for further evaluation. We show here that this synergy observed both in vitro and in vivo results from multiple convergent pathways to include increased caspase activation, HDAC inhibitor-mediated TNF-α upregulation, and alternative NF-kB signaling. These findings provide a rationale for the integration of SMAC mimetics and HDAC inhibitors in clinical trials for recurrent ovarian cancer where treatment options are still limited.

## 1. Introduction

Ovarian cancer is the 5th most lethal malignancy for women; 13,980 women died from ovarian cancer last year in the United States [1]. The US Surveillance, Epidemiology, and End Results (SEER) database reports that 5-year overall survival in patients with stage III and stage IV cancer is 25%. While the current standard treatment of ovarian cancer is cytoreductive surgery in conjunction with platinum plus taxane-based chemotherapy, the mortality of ovarian cancer has only slightly improved in the past decade [2]. In general, ovarian cancers show an initial response to chemotherapy but gain anti-apoptotic ability by increasing the expression of anti-apoptotic molecules or inactivating pro-apoptotic cell death components, leading to tumor recurrence and drug resistance [3]. In this regard, various attempts to restore the apoptotic mechanism of cancer cells are ongoing [4].

Apoptosis is the most important regulated mechanism of cell death, led by two distinct pathways, the death receptor signal pathway (extrinsic apoptosis) and the mitochondrial pathway (intrinsic apoptosis), both result in the activation of caspases. To prevent inappropriate activation of caspases, apoptosis is strictly regulated by both pro-apoptotic and anti-apoptotic proteins. One such protein is receptor-interacting protein 1 (RIP1), which is ubiquitinated by two cellular inhibitor of apoptosis proteins (cIAPs), cIAP1 and cIAP2 [5]. The ubiquitinated RIP1 complex leads to downstream signaling, resulting in the transcription of NF-kB gene targets. A second mitochondrial activator of caspases (SMAC) proteins released from mitochondria ubiquitinate XIAP to promote apoptosis [6], but this mitochondrial pathway is blocked by high expression levels of IAP proteins in some chemotherapy-resistant recurrent cancers. In ovarian cancer patients, high cIAP1/2 expression correlated with significantly shorter survival compared to patients with low cIAP1/2 expressing cancers in multivariate analysis [7].

Small molecules designed to mimic the IAP binding motif of SMAC, which binds to the BIR (baculoviral IAP repeat) 2 and BIR3 domains of cIAPs, are called SMAC mimetics (SMs) [8]. After SMs remove cIAPs by auto-ubiquitination, de-ubiquitinated RIP1 can form distinct death complexes with tumor necrosis factor receptor type 1-associated death domain protein (TRADD), resulting in tumor necrosis factor-α (TNF-α) induced cell death through caspase-mediated apoptosis or RIP3/MLKL mediated necroptosis. SMs are intended to treat cancer with minimal toxicity to normal cells since they preferentially target cancer cells which depend on cIAPs more so than healthy cells. To date, SMs have entered clinical trials for hematological and solid cancers [9].

We previously completed a phase 2 clinical study of the SMAC mimetic, birinapant, in women with relapsed platinum-resistant or platinum-refractory epithelial ovarian cancer. Birinapant showed consistent on-target suppression of cIAP1 in tumor biopsies and peripheral blood mononuclear cells, but single-agent anti-tumor activity did not meet the pre-specified primary endpoint of 20% response rate in order to justify proceeding with the development of the single agent [10]. Because of the strong on-target effect but minimal efficacy, we undertook a drug screening to find rational combination therapies to enhance the anti-cancer activity of SMAC mimetics. Matrix drug screening using a library of 1912 compounds crossed with birinapant found that histone deacetylase (HDAC) inhibitors (entinostat, panobinostat, vorinostat, romidepsin, and Trichostatin A) are highly synergistic with SMs in killing ovarian cancer cells [11]. Here, we tested the hypothesis that HDAC inhibitors sensitize cancer cells to SMs, suggesting that the combination of SM and HDAC inhibitors can be a novel anti-cancer therapy.

Histone deacetylases (HDACs) are important epigenetic regulators of gene expression, removing histone acetylation enzymatically. There are three main HDAC classes (I, II, and IV) comprised of at least 11 HDACs. Histone acetylation is generally associated with elevated gene transcription, but overexpression of HDACs in cancer is thought to repress tumor suppressor genes by histone deacetylation, thus resulting in tumor progression. With this goal, various HDAC inhibitors are in clinical trials as novel cancer therapeutics [12].

In this study, we demonstrate a mechanism of synergy between SMs and HDAC inhibitors that involves TNF-α. Recently, a genome-wide siRNA screen identified that the regulation of TNF-α mRNA expression by transcription factor SP3 is a critical factor for SMs mediated cancer cell death [13]. TNF-α is a trigger of the death receptor signal, which is necessary for immunity and anti-cancer effects through regulated cell death, especially in the presence of SM-facilitated apoptosis. We, therefore, hypothesized that HDAC inhibitors sensitize cancer cells to SMAC mimetics by enhancing TNF-α production via SP3. Here, we tested both birinapant, a dimeric peptidomimetic compound, and another SMAC mimetic, tolinapant, a non-peptidomimetic antagonist of cIAPs and XIAP, discovered using fragment-based drug design [14]. These SMs were combined with different HDAC inhibitors, including the class I selective HDAC inhibitor, entinostat, reported to have an anti-tumor effect in an ovarian cancer model [15], the broader-acting panobinostat, as well as romidepsin and vorinostat. We demonstrate that the synergy is not restricted to individual drugs but is a class effect of the SMs and HDAC inhibitors. By identifying the mechanism of this synergy, we hope to move this combination therapy forward clinically to present a novel treatment option for women with chemo-refractory ovarian cancer.

## 2. Materials and Methods

### 2.1. Cell Culture

Ovarian cancer cell lines were obtained from the NCI-Frederick Developmental Therapeutics Program tumor/cell line repository (Frederick, MD, USA). Human ovarian cancer cell lines (OVCAR3, OVCAR8, SKOV3, TOV21G, ES-2, A2780, and IGROV1) were grown in RPMI medium supplemented with 10% FBS, 1% penicillin/streptomycin. Mouse ovarian cancer cell line ID8 was grown in DMEM medium with 4% FBS, 1% penicillin/streptomycin, insulin (5 μg/mL), transferrin (5 μg/mL), sodium selenite (5 ng/mL) (ITX mix, Sigma -Aldrich #I-1884, St. Louis, MO, USA). All cells were maintained at 37 °C under a 5% CO_2_ atmosphere. Cell lines were authenticated via Short Tandem Repeat at Frederick National Laboratory. Authenticity was confirmed against the ATCC database (www.atcc.org/CulturesandProducts/CellBiology/STRProfileDatabase/tabid/174/Default.aspx (accessed on 1 May 2019)), CLIMA database (http://bioinformatics.istge.it/clima/ (accessed on 1 May 2019)), and NCI-60 database published data.

### 2.2. XTT Viability Assay

Further in vitro testing verified synergistic activity between tolinapant and entinostat, romidepsin, or vorinostat. OVCAR3, OVCAR8, SKOV3, TOV21G, ES-2, A2780, IGROV1, and ID8 cancer cell lines were seeded in 96-well plates at a density of 2–4 × 10^3^ cells/well and incubated for 24 h prior to drug addition. In the case of dual drug treatment, both drugs were added simultaneously. Then, 72 h after drug application, XTT-PMS dye was added to each well, incubated for 1.5 h, and then OD 450 nm was measured to determine cell viability (Molecular Devices SpectraMax ID3 plate reader).

### 2.3. Western Blot

OVCAR3, OVCAR8, SKOV3, TOV21G, ES-2, A2780, IGROV1, and ID8 cells were plated in 6-well plates at a density of 1.0 × 10^6^ to allow for evaluation of protein expression following treatment with single agent tolinapant, entinostat alone, or combination therapy. Cells were treated for 24 h or 72 h, and then total protein was extracted using M-PER buffer (Thermo Scientific, Waltham, MA, USA) according to the manufacturer’s protocol. Concentrations were estimated with the BCA Protein Assay Kit (Thermo Scientific). SDS-Page and Western analysis were performed using the NuPage system (Invitrogen, Waltham, MA, USA) and the Supersignal Chemiluminescent Substrate System (Thermo Scientific), respectively. The following primary antibodies were used: GAPDH (MAB374, Sigma-Aldrich), SMAC (#2954S CST), DR5 (#8074S CST), Acetyl-Histone H3 (K9/K14, #9677S CST), HDAC1 (#34589 CST), p52 (MAB05-361 Sigma-Aldrich), XIAP (#2042S CST), human cIAP1 (AF8181 R&D), mouse cIAP1 (ALX-803-335-C100 ENZO), SP3 (sc-365220 Santa Cruz), Phospho-NF-kB p65 (#3033 CST), Cleaved Caspase3 (#9664 CST). Protein signal quantitation used Image Studio Lite Ver 5.2 (LI-COR). Original blots can be found at Appendix A.

### 2.4. Co-Immunoprecipitation (Co-IP)

OVCAR3, OVCAR8, and TOV21G cells were treated with single-agent tolinapant, entinostat alone, or combination therapy for 24 h, and total protein was extracted using RIPA buffer (Thermo Scientific) according to the manufacturer’s protocol, followed by sonication (Fisherbrand™ Model 120 Sonic Dismembrator). Concentrations were estimated with the BCA Protein Assay Kit (Thermo Scientific). A total of 1 μg of total cellular protein and 1 μg of primary antibodies are incubated overnight at 4 degrees with protein A/G beads (sc-2003 Santa Cruz). Beads were washed and boiled in SDS sample buffer containing DTT before analysis of the eluted proteins by Western blotting. Western blots of the same lysate before immunoprecipitation were used for comparison.

### 2.5. Caspase Activity Assay

Caspase 3/7, Caspase 8, and Caspase 9 activity were measured in OVCAR3 and OVCAR8 cells lines using Caspase-Glo luminescence assays (Promega, G8091, G8201, G8211) according to the manufacturer’s specifications after exposing cells to panobinostat 10 nM alone and in combination with birinapant 20 μM for 24 h. All drug exposures occurred with or without 10 ng/mL of TNF-α. Activity data were normalized to viable cell number and measured in an identical plate by XTT assay as described.

### 2.6. NF-kB Reporter Assay

OVCAR3 and OVCAR8 cell lines were selected given their high canonical NF-kB activity. Cells were transduced with a luciferase reporter to measure NF-kB (p65) activation. Reporter cells were then plated at a density of 2000 cells/50 μL per well in a 96-well plate and then exposed to entinostat alone and in combination with tolinapant for 24 h. All drug exposures occurred with and without TNF-α (10 ng/mL). Quantitative luminescence was measured by Promega Luciferase Assay using a Molecular Devices SpectraMax ID3 plate reader and normalized to a cellular viability assay using XTT-PMS dye.

### 2.7. RNAi Experiments

OVCAR8 cells were transfected with siRNAs using DharmaFECT 1 Transfection Reagent per standard manufacture procedure (Horizon Discovery), and drugs were added 2 days later. The siRNAs used were as follows: ON-TARGET plus Non-targeting Pool (D-001810-10; Horizon Discovery, Waterbeach, UK), ON-TARGET plus siRNA against human TNF-α (L-010546; Horizon Discovery).

### 2.8. Cytokine Assay

Secreted cytokine levels of IL-6, IL-8, and TNF-α were measured in culture supernatants using the Mesoscale multiplex assay after treating OVCAR3 and OVCAR8 cells overnight with Panobinostat 20 μM alone and in combination with birinapant 10 nM per manufacturer’s protocol (MesoScale Discovery, Rockville, MD, USA).

### 2.9. RNA Extraction and Quantitative Real-Time PCR

Total RNA was isolated using RNeasy Mini Kit (Qiagen, Hilden, Germany) per the manufacturer’s protocol. The final RNA concentration was determined with a NanoDrop spectrophotometer using the 260/280 absorbance ratio. Total purified RNA was reverse transcribed with random primers using High-Capacity cDNA Reverse Transcription Kit (Applied Biosystems, Waltham, MA, USA) per the manufacturer’s protocol. The resulting cDNA was used as a template for quantitative real-time PCR (qRT-PCR). Analysis of gene expression was performed on ViiA7 Real-time PCR System (Applied Biosystems) using TaqMan probe assays with GAPDH and ACTB as control. Quantitation and normalization of relative gene expression were accomplished using ddCT (Delta-Delta-cycle threshold) method. Catalog numbers for commercial primers are provided in Appendix A.

### 2.10. Chromatin Immunoprecipitation (ChIP)—qPCR Assay

The SimpleChIP Enzymatic Chromatin IP Kit (magnetic beads) was purchased from Cell Signaling Technology, and assays were performed according to the manufacturer’s instructions. Antibodies for RelA (NF-kB p65 #8242S CST), p50 (NF-kBp105/50 #13586S), RelB (#10544 CST), p52 (NF-kB p100/52 #37359S CST), Acetyl-Histone H3 (K27, #4353S CST), and SP3 (sc-365220 Santa Cruz). Genome browser (https://genome.ucsc.edu/index.html (accessed on 1 August 2019)) and Jasper (http://jaspar.genereg.net/ (accessed on 1 August 2019)) were used to evaluate DNA sequences for transcription factor binding sites. Analysis of ChIP was performed on a ViiA7 Real-time PCR System (Applied Biosystems) using QuantiTect SYBR Green PCR Kit (Qiagen). The quantification of transcription factors binding to target sites was calculated by measuring the ddCT (Delta-Delta-cycle threshold) ratio of chromatin immunoprecipitation (ChIP) to 2% Input, and the normal rabbit IgG antibody served as a negative control. All primers used for ChIP PCRs are listed in Appendix A.

### 2.11. In Vivo Mouse Studies

#### 2.11.1. Xenograft Model

1–2 × 10^6^ of OVCAR8 cells were counted and prepared as suspensions in 0.5 mL PBS for subcutaneous (flank) injections into 6–8 weeks old athymic nude female mice. Tumors were grown for two weeks before the mice were randomized into treatment groups. Mice then received intraperitoneal (IP) treatment with vehicle control (5% dextrose), Panobinostat 10 mg/kg, per oral (PO) treatment of tolinapant 16 mg/kg, or combination panobinostat plus tolinapant. Body weights and tumor measurements were taken twice weekly for 8–10 weeks or as required by humane endpoints. Subcutaneous tumor volumes were calculated according to the formula V = 1/2(length × width^2^).

#### 2.11.2. Survival Study: Immune Deficient Model

1–2 × 10^6^ OVCAR8 cells were injected IP into 6–8 weeks old athymic nude female mice and allowed to grow for 2 weeks before IP treatment with vehicle control (5% dextrose), entinostat 20 mg/kg, tolinapant 10 mg/kg, or combination entinostat plus tolinapant. Mice were followed until euthanasia endpoints and scored for overall survival. Animal care was provided in accordance with procedures in the Guide for the Care and Use of Laboratory Animals. Experiments were carried out according to a protocol approved by the NCI Animal Care and Use Committee.

#### 2.11.3. Survival Study: Immune Competent Model

2 × 10^6^ ID8 p53−/− cells were injected IP into 6–8 weeks old C57/BL6 female mice and allowed to grow for 4 weeks before IP treatment with vehicle control (5% dextrose), entinostat 20 mg/kg, tolinapant 10 mg/kg, or combination entinostat plus tolinapant with 200 μg/mouse of control IgG or anti-mouse PD1 antibody (#29F.1A12, Bio X CellX, Lebanon, NH). ID8 p53−/− cells were kindly provided by Josephine Walton [16]. Mice were followed until euthanasia endpoints and scored for overall survival. Animal care was provided in accordance with procedures in the Guide for the Care and Use of Laboratory Animals. Experiments were carried out according to a protocol approved by the NCI Animal Care and Use Committee.

### 2.12. Cytokine Assay in Mouse Ascites

The 2 × 10^6^ ID8 p53−/− cells were injected IP into 6–8 weeks old C57/BL6 female mice and allowed to grow for 5 weeks before IP treatment with vehicle control (5% dextrose), entinostat 20 mg/kg, tolinapant 10 mg/kg, or combination entinostat plus tolinapant. Ascites were collected after 24 h from final treatment, and TNF-α were measured using Quantikine ELISA Mouse TNF-α Immunoassay kit per the manufacturer’s protocol (R&D Systems, Minneapolis, MN, USA).

### 2.13. Statistical Analysis

All in vitro experiments were conducted in duplicate or triplicate for each experimental condition, as noted above. Results were analyzed for statistically significant differences using two-tailed *t*-tests (2 groups) or ANOVA multiple comparison tests (3 or more groups) in GraphPad Prism version 8.0 for Windows (GraphPad Software, San Diego, CA, USA, www.graphpad.com (accessed on 1 November 2019)). Overall survival was estimated using the Kaplan–Meier method, with differences between treatment groups evaluated using a long-rank test in GraphPad Prism version 8.0 for Windows; *p* values < 0.05 were considered statistically significant. Synergistic effects on XTT cell viability assay between two drugs are analyzed with Combenefit (https://www.cruk.cam.ac.uk/research-groups/jodrell-group/combenefit (accessed on 1 November 2019)) [17]. Combination indexes were analyzed with CalcuSyn Version 2.0 (http://www.biosoft.com/w/calcusyn.htm (accessed on 1 November 2019)). Quantification of Western blots was performed with densitometry using LI-COR Image Studio Lite version 5.2 (LI-COR Biotechnology, Lincoln, NE, USA).

### 2.14. Multiplexed Immuno-Fluorescence Imaging Using CODEX

#### 2.14.1. Single Cell Analysis

CODEX platform [18] was used to evaluate the impact of tolinapant, entinostat, and their combination on the immune tumor microenvironment (iTME) and to assess whether combination therapy may be a new treatment option in ovarian cancer. Multiplexed Immuno-Fluorescence Imaging was used as a powerful tool to evaluate iTME reproducibly. Multiplexed imaging technologies allow capturing many parameters of single cells while preserving their spatial location. We randomized a total of 20 C57B6 mice, each with tumor cells (ID8 p53KO) implanted in their ovary, into groups #1–5 as Control, #6–10 as the tolinapant group, #11–15 as the entinostat group, and #16–20 as the combination treatment group. After drug treatment, ovarian tumor resection was performed, and appropriate specimens were prepared for Multiplexed Immuno-Fluorescence Imaging. We acquired and processed CODEX images for optimal performance in HALO image analysis. With HALO algorithms and modules, CODEX multiplexed IF images from mice ovarian tumor tissue were generated using 18 barcoded antibodies. The major structures within ovarian cancer were imaged, such as the cortex, medulla, follicle, and fallopian tube. We evaluated tissue morphology based on Hoechst nuclear staining. The various settings for identifying Hoechst are as follows.: Nuclear Contrast Threshold 0.5, Minimum Nuclear Intensity 0.1, Maximum Image Brightness 1, and Nuclear Segmentation Aggressiveness 0.65. After that, we set the weak, moderate, and strong thresholds for all dyes other than Hoechst, respectively, and we counted the relative distributions of each cell in the ovarian tumor. The key tumor cell and primary T cell types could be visualized and compared in mouse ovarian tumor tissue. Specifically, the Helper T cell, cytotoxic T lymphocyte (CTL), regulatory T cell (Treg), and PD-1 positive T cell were identified.

#### 2.14.2. Spatial Plot Analysis

After counting the number of each cell, the localization of each cell was identified by Spatial Plot. To evaluate the effects of various treatments on ovarian tumor tissue, we needed to distinguish tumor cells from the rest of the tumor within the ovarian tumor tissue. After identifying the localization of tumor cells, we created a Density Heatmap in Spatial Analysis to evaluate the density of tumor cells. A tumor annotation line (yellow line) was drawn for each sample using the Density Heatmap to determine the tumor site and other sites with a high degree of accuracy. The number of various cells was then counted in the tumor and other areas and analyzed using one-way ANOVA, showing that T cells (red) tend to cluster around the tumor cells (light blue) (Appendix A). A representative figure showing the relationship between the outer edge of the sample (black line), the tumor annotation line (yellow line), and the various T cells (blue: Helper T cells, green: Tregs, yellow: CTLs) is shown in Appendix A.

## 3. Results

### 3.1. Tolinapant Is Synergistic in Combination with HDAC Inhibitors in Killing Ovarian Cancer Cells

To measure the efficacy of tolinapant in vitro, we performed a dose titration of tolinapant and measured cell viability by XTT assay in 9 ovarian cancer cell lines (OVCAR3, OVCAR4, OVCAR5, OVCAR8, CAOV4, OV90, PEO1, PEO4, SKOV3) representing different histological subtypes of ovarian cancer. Tolinapant was found to have variable single-agent activity, with the lowest IC50 in SKOV3, at a clinically achievable concentration (tolinapant IC50 < 50 μM). In the 9 tested cell lines, OVCAR3 was the most resistant to tolinapant (Figure 1A). Although OVCAR3 was resistant to the SMAC mimetic tolinapant as a single agent, the addition of exogenous TNF-α (10 ng/mL) sensitized OVCAR3 to 72 h exposure to tolinapant (Figure 1B). TNF-α itself did not affect cell viability.

Our previously published matrix drug screen tested a library of 1912 compounds crossed with the SM birinapant and found that the class of HDAC inhibitors (entinostat, panobinostat, vorinostat, romidepsin, and Trichostatin A) are highly synergistic in killing ovarian cancer cells [11]. Consistent with our screening results with birinapant, the newer, orally available SM tolinapant also shows a synergistic effect with the class I HDAC inhibitor, entinostat (Figure 1C). The synergy was stronger with higher tolinapant concentration (tolinapant 50 μM > 25 μM > 12.5 μM). We confirmed the effect in an additional cell line, OVCAR8, representing high-grade serous ovarian cancer (HGSOC), the most common histologic subtype, and the calculated combination index (CI) was below 1 for each cell line, indicating synergy (Figure 1D).

The cell line SKOV3, most sensitive to tolinapant as a single agent, is not a typical high-grade serous ovarian cancer (HGSOC) cell line. It is more likely to be endometrioid, emerging from endometriosis-associated cancer, with loss of ARID1A function. We, therefore, tested the sensitivity of additional ARID1A-mutant or endometriosis-related ovarian cancer cell lines to tolinapant and entinostat to evaluate whether these characteristics might serve as markers for sensitivity to SMAC mimetics. We also included a mouse cell line (ID8). All tested cell lines (A2780, TOV21G, SKOV3, ES2, IGROV1, and ID8) showed a synergistic effect between tolinapant and entinostat (Appendix A). Cell viabilities of combination therapy were significantly lower than either tolinapant or HDAC inhibitor single treatment, and the calculated combination index (CI) was again below 1 for each cell line, indicating synergy. Interestingly, the TOV21G (ARID1A mutant), IGROV1 (ARID1A mutant), and ES2 (clear cell subtype, potentially endometriosis-related) cell lines had varying sensitivity to tolinapant single agent. We proceeded to confirm that the synergy is a class effect of all HDAC inhibitors and not restricted to only entinostat. The combination of tolinapant with romidepsin or vorinostat was tested in four of the ovarian cancer cell lines representing different histologic subtypes (Appendix A). Each HDAC inhibitor was titrated in matrix format with the SMAC mimetic in order to calculate the most synergistic concentration for each drug. Once again, synergy was evident based on achieving a calculated combination index below 1.

### 3.2. Tolinapant Inhibits cIAP1 and XIAP, and Entinostat Acetylates Histone H3

Next, we assessed in vitro the on-target effects of tolinapant and entinostat in OVCAR3 and OVCAR8 cell lines. These cell lines were chosen since they represent high-grade serous ovarian cancer, the most common histologic subtype. They also represent chemo-resistant recurrent ovarian cancer, which is the most significant cause of morbidity and mortality from ovarian cancer. Tolinapant treatment depleted cIAP1 protein both alone and in combination (Figure 2A). In the same way, entinostat treatment increased acetylated Histone H3 (Figure 2B). We sought to understand how tolinapant affects the interaction between SMAC and XIAP. Tolinapant did not change the protein level of either XIAP or SMAC (Figure 2C). In the absence of tolinapant, SMAC was co-immunoprecipitated with XIAP, but in the presence of tolinapant, the interaction between SMAC and XIAP was completely inhibited in OVCAR3 and OVCAR8 cells (Figure 2C). This loss of cIAP1 combined with releasing SMAC from suppression by XIAP is essential for total activation of apoptosis and a sustained pro-apoptotic effect downstream of TNF-α signaling.

### 3.3. HDAC Inhibitors and SMs Synergistically Increase Cell Death through NF-kB Activation

We sought to validate the class effects of HDAC inhibitors and SMAC mimetics in ovarian cancer cell lines. The SMAC mimetic birinapant is a peptidomimetic dimer that induces IAP degradation, and panobinostat is a pan-HDAC inhibitor. Caspase 3/7, 8, 9 activation was quantified by luminescence assay as a measure of apoptotic activity. Caspase activation was quantified in OVCAR3 and OVCAR8 cells with relative luminescence (Appendix A). All luminescence data were normalized to cell viability as determined by XTT assay performed in parallel. Experiments were performed both with and without added TNF-α (10 ng/mL) to represent external TNF-α activation in the tumor microenvironment. Caspase 3/7 was measured as the final common pathway of apoptosis, Caspase 8 as extrinsic apoptosis, and Caspase 9 as intrinsic apoptosis. As expected, the combination of birinapant and panobinostat increased the activation of caspase 3/7 greater than each single agent. Caspase 9 was similarly activated as caspase 3/7, but caspase 8 was highly variable and did not reach statistical significance.

Since IAP proteins are known to modulate classical alternative NF-kB signaling [19], we quantified NF-kB activation using NF-kB luciferase reporter lines. NF-kB luciferase reporter lines were established in OVCAR3 and OVCAR8 cell lines. Reporter cell lines were treated with birinapant or panobinostat alone or in combination. All luminescence data were normalized to cell viability as determined by XTT assay performed simultaneously. Based on the NF-kB consensus sequence in the reporter construct, we were unable to distinguish between classical or alternative NF-kB pathways from these results.

Experiments were performed both with and without external TNF-α (10 ng/mL) to specifically activate the classical pathway of NF-kB signaling since it would not affect the alternative pathway. Panobinostat appeared to activate NF-kB signaling when added by itself or in combination, irrespective of the addition of TNF-α (Appendix A). Interestingly, birinapant resulted in a small amount of activated NF-kB reporter when added by itself or in combination in the absence of TNF-α but attenuated the panobinostat-induced NF-kB activation in the presence of TNF-α (Appendix A). Similarly, entinostat alone and the combination of entinostat with tolinapant activated NF-kB function in the absence of TNF-α (Figure 3A). Here, the tolinapant attenuated the NF-kB activation by both TNF-*α* alone and by entinostat in the presence of TNF-*α*. These results suggest that SMs may be activating alternative NF-kB signaling but blocking classical NF-kB activity triggered by TNF-α stimulation.

To confirm the changes at the protein level, we performed a Western blot after cells were exposed to single agents and a combination of SMAC mimetic tolinapant and HDAC inhibitor entinostat in OVCAR8 and TOV21G cells (Figure 3B and Appendix A). Tolinapant alone or in combination with entinostat showed an increased presence of alternative NF-kB isoform p52. The cIAP1 is known to destabilize NF-kB-inducing kinase (NIK). In the absence of cIAP1, stabilized NIK can process p100 to p52, resulting in NK-kB alternative pathway activation [20]. This effect on the alternative NF-kB pathway was related to tolinapant, likely due to its mechanism of inducing cIAP1 degradation. Interestingly, cleaved caspase 3 showed a similar pattern related to tolinapant, suggesting caspase activation by the release of SMAC from XIAP inhibition. The classical NF-kB isoform p65, also known as RelA (v-rel reticuloendotheliosis viral oncogene homolog A), was phosphorylated synergistically after tolinapant and entinostat treatments. Phosphorylation of RelA plays a key role in NF-kB activation and suggests the existence of activated TNF-α signaling. This pattern, however, did not follow that of the NF-kB reporter activity. Finally, PARP was cleaved after tolinapant treatment, most strongly in combination, suggesting synergistic induction of apoptosis.

### 3.4. HDAC Inhibitor-Mediated TNF-α Secretion in Tumor Cells Is Critical for Synergy with SMs

Based on the pattern of activation of NF-kB and phosphorylation of RelA, we hypothesized that TNF-α autocrine signaling is enhanced by SMs and HDAC inhibitors. The mRNA levels of TNF-α were measured after tolinapant alone or in combination with entinostat (Figure 4A and Appendix A). Entinostat increased TNF-α-mRNA in all tested cell lines, which was further increased by the combination. Interestingly, tolinapant single treatment did not significantly affect TNF-α-mRNA in OVCAR3 and OVCAR8 (Figure 4A) but caused a measurable increased TNF-α-mRNA in TOV21G and SKOV3 (Appendix A). This suggests that the partial sensitivity of TOV21G and SKOV3 to tolinapant single treatment may be related to the ability of tolinapant to increase TNF-α secretion in these cell lines.

The combination effect of the HDAC inhibitor and SMAC mimetic was confirmed at the protein level. Secreted cytokine levels of IL-1ß, IL-6, IL-8, and TNF-α were measured in culture supernatants after treating cells with birinapant alone, panobinostat alone, or both drugs in combination. The production of IL-1ß, IL-6, and IL-8 was minimally varied across treatment groups and did not reach statistical significance. There was, however, a notable and statistically significant increase in TNF-α production after the treatment of cells with a single-agent drug that further increased with the combination (Appendix A).

To further demonstrate the importance of autocrine TNF-α in the synergistic anti-tumor effect of the combination of SM and HDAC inhibitor, TNF-α-mRNA was knocked down with siRNA in OVCAR3 and OVCAR8, cell lines that are not sensitive to single agent SM and upregulate TNF-*α* only with the combined drugs. Cells transduced with si-TNF-α or si-Control were exposed to tolinapant with or without entinostat. Importantly, knockdown of TNF-α with RNA interference prevented the synergistic anti-tumor effect of tolinapant combined with entinostat (Figure 4B). This result suggests that HDAC inhibitor-mediated TNF-α plays a critical role in the synergistic anti-tumor effect.

### 3.5. Entinostat Enhances the Production of TNF-α through Acetylation of TNF-α Promoter Region

In response to TNF-α, SMs relieve RIP1 from cIAP-mediated ubiquitination, promoting the formation of complexes that can activate survival signaling through the classical NF-kB pathway [21]. Antagonism and subsequent depletion of the cIAPs lead to the stabilization of NIK, which activates the alternative NF-kB pathway [22]. In this setting, the HDAC inhibitor could play a critical role in upregulating TNF-α transcription for autocrine secretion. We hypothesized that HDAC inhibitors might enhance TNF-α transcription by increasing the amount of histone acetylation at the TNF-α promoter region. To demonstrate the mechanism of upregulated transcription of TNF-α, Chromatin-IP (ChIP) qPCR was performed in OVCAR8 cells exposed to single agents and the combination of tolinapant and entinostat.

Both NF-kB and SP3 are critical transcription factors for TNF-α [13]. There are three predicted NF-kB consensus response elements and six predicted SP3 binding sites closely associated with the TNF-α promoter region. We defined the three predicted NF-kB consensus response elements as NFkB1 (−873 bp from start codon), NFkB2 (−612 bp), and NFkB3 (−97 bp). In the same way, the six predicted SP3 binding cites are defined as SP3_1 (−1010 bp), SP3_2 (−869 bp), SP3_3 (−647 bp), SP3_4 (−550 bp), SP3_5 (−174 bp), and SP3_4 (−54 bp) (Figure 5A). First, we evaluated histone acetylation using an anti-acetylated histone H3 (AcH3) antibody to immunoprecipitate the chromatin region. Entinostat significantly increased AcH3 in the region designated NFkB2, suggesting that entinostat allows acetylation of histone H3 to persist in selective regions of the TNF-α promoter region (Figure 5B).

We next evaluated the DNA binding of four NF-kB proteins (RelA, p50, RelB, and p52) and the SP3 protein. Although the low DNA binding affinities of these proteins impaired statistical significance, all treatments tended to upregulate the DNA binding of NF-kB in the NFkB2 region (Figure 5C). Combination therapy had a tendency to enhance p52 to bind all three DNA binding sites, suggesting the importance of the NF-kB alternative pathway in TNF-α autocrine signaling. In a similar way, all treatments appeared to upregulate the DNA binding of SP3 on the SP3_4 region (Figure 5D). SP3 protein levels remained unchanged by treatment (Appendix A). These results suggest that entinostat enhances transcription factor binding in the TNF-α promoter region by site-specific histone acetylation in the SP3_4 region and NFkB2 regions, which are closely arranged in a region is thought to play an important role in the treatment-induced TNF-α autocrine (Figure 5E).

### 3.6. HDAC Inhibitors Have Synergistic Anti-Tumor Effect with Tolinapant In Vivo

We next evaluated the effects of combining tolinapant and HDAC inhibitors in mouse models of ovarian cancer. Both subcutaneous and intraperitoneal models were investigated. The intraperitoneal model closely mimics the pattern of recurrent ovarian cancer in humans, which makes this model ideal for measuring changes in overall survival. The intraperitoneal model, however, is not well suited for measuring tumor volume due to the widespread dissemination. We used the subcutaneous model to measure changes in tumor volume more accurately. OVCAR8 xenografts were established using subcutaneous injection into 6–8-week-old athymic female mice. Tumors were grown for 2 weeks until they reached an average volume of 50–100 mm^3^, at which time mice were randomized into groups of 5 for treatment. Mice received 3 weeks of vehicle, single agent tolinapant (16 mg/kg, PO), panobinostat (10 mg/kg, IP), or combination treatments. Body weights and tumor measurements were taken twice weekly for 8–10 weeks, and subcutaneous tumor volume was calculated. The average tumor volume was lower for mice treated with single-agent tolinapant or panobinostat compared to vehicle-treated mice and lowest for mice receiving combined treatment (Figure 6A).

In a separate experiment, overall survival was assessed using an orthotopic intraperitoneal model of OVCAR8 in 6–8-week-old athymic female mice. After 2 weeks of xenograft growth, mice were treated with 3 weekly IP treatments of single or combined tolinapant 10 mg/kg and entinostat 20 mg/kg. Overall survival was dramatically increased in mice treated with combination therapy (*p* < 0.0001), with a median survival of 60 days (vehicle; 36 days, tolinapant; 32 days, and entinostat: 37 days). Mice treated with single-agent therapy had a median survival of approximately 51 days (Figure 6B).

An immune-competent mouse model was examined in order to assess the effects of these drugs on the tumor microenvironment. ID8-p53KO mouse ovarian cancer cell line was injected into the ovarian bursa of C57B6 mice. Tumors were allowed to grow for 4 weeks, and then mice were treated with 1 week of vehicle, single drugs, or a combination of tolinapant and entinostat. Mice were euthanized, and ovaries were harvested for analyses (Figure 6C). TNF-α was measured in mouse ascites with ELISA and found to be significantly increased in mice treated with entinostat compared to control (*p* < 0.01) and further increased by the combination of entinostat and tolinapant (*p* < 0.001). This is consistent with our findings in in vitro assays (Figure 4B) and highlights the role of TNF-α in the synergistic activity of this combination.

Quantitative analysis of immune cell infiltrates was performed using the CODEX platform [18]. Each mouse ovary was stained with 18 markers to identify subsets of immune cells (Figure 7A and Appendix A). Differences between treatment groups were found in T cell subsets, with significantly decreased total T cells, helper T cells, and T regulatory cells, but not in cytotoxic T lymphocytes (CTL) in the combined treatment group compared to vehicle-treated mice (*p* = 0.04, 0.008 and 0.007, respectively) (Figure 7B). Interestingly, PD1 expression was significantly increased in the T helper cell population in the combination treatment group compared to vehicle-treated and entinostat-treated mice (*p* = 0.004 and *p* = 0.003, respectively) (Figure 7C). Because of the increased PD1 expression on the T helper cells, we hypothesized that combining anti-PD1 antibody with tolinapant and entinostat would improve outcomes in the immune-competent mice. We, therefore, proceeded with the intraperitoneal inoculation of ID8-p53KO into immune-competent C57B6 mice and treated 8 groups of 10 mice with distinct combinations of treatments (Figure 7D). Consistent with our hypothesis, mice treated with the combination of tolinapant, entinostat, and anti-PD1 antibody had the longest median survival, and this was statistically significantly improved over all other groups, particularly the group treated with tolinapant, entinostat, and IgG control antibody (*p* = 0.003) (Figure 7D).

## 4. Discussion

IAP inhibition with SMs is a promising anti-cancer treatment option but has limited efficacy as a single agent. In this study, we demonstrated improved efficacy with the combination of SMs and HDAC inhibitors, a synergistic combination that was discovered through our global unbiased matrix drug screen that we previously reported [11]. Importantly, we demonstrated this synergy across multiple cell lines, and using different SMs and HDAC inhibitors, confirming that this is a class effect and not specific to individual targeted agents. Our data support a mechanism of synergy whereby the HDAC inhibition increases transcription of TNF-α, and the IAP inhibition by SMAC mimetic shunts TNF receptor signaling to activate caspase cleavage and subsequent apoptosis (Figure 8).

Some ovarian cancer cell lines, such as SKOV3, TOV21G, IGROV1, and ES2, are susceptible to single-agent SM. These cell lines have some common features which might cause synergism with SMs, such as the secretion of TNF-α. In addition, SKOV3, TOV21G, and IGROV1 have *ARID1A* mutation, and TOV21G and ES2 are clear cell subtypes [23]. In contrast to most high-grade serous ovarian cancer, which has *TP53* mutation [24], many (57–67%) of ovarian clear cell cancer (OCCC) have *ARID1A* mutation [25,26]. *ARID1A* mutant ovarian cancers may be related to OCCC because *TP53* and *ARID1A* are almost mutually exclusive in ovarian cancer. OCCC is the most refractory to standard chemotherapeutic regimens (platinum and taxane) among epithelial ovarian cancers, and the median survival of women with stage III/IV OCCC is significantly lower than those with HGS [27]. Additionally, OCCC arises from endometriosis [28]. Endometriosis is difficult to cure with current hormonal therapy, but IAP inhibition has demonstrated a positive effect in controlling endometriosis in a mouse model [29,30]. Endometriosis lesions highly express cIAPs [28] which could contribute to the generation of endometriosis but also provide a target to prevent or treat future OCCC. Endometriosis-related ovarian cancer might have a higher sensitivity to SMAC mimetics because those cancers have a greater tendency to express TNF-α compared with high-grade serous ovarian cancer. Previous work showed that endometrioid ovarian cancers have a higher positive rate of TNF-α (83.3%, 10/12 cases) compared with high-grade serous ovarian cancer (40%, 20/50 cases) on immunohistochemistry staining [31]. Therefore, SMs could be developed as a treatment option for endometriosis or OCCC.

In the presence of TNF-α, the SMs had a heightened anti-tumor effect. Mechanistically, the identified upregulation of TNF-α provides a stimulus for apoptosis that is augmented by the presence of the SMAC mimetic. Without the SM, the TNF-α in the presence of cIAP1 can stimulate classical NF-kB signaling and block apoptosis. The SM depletes cIAP1, and thus, TNF receptor signaling leads to caspase activation and apoptosis [32]. Our results show that the HDAC inhibitor increases TNF-α mRNA and protein, and these are further increased with the combination. SMs-mediated cIAP1 depletion combined with caspases released from XIAP is essential for full activation of apoptosis downstream of the TNF-α signaling pathway. Cell lines OVCAR3 and OVCAR8 showed maximal activation of Caspase 3/7 and 9 with the combination of SMAC mimetic and HDAC inhibitor in the presence of TNF-a. OVCAR8 showed less reactivity to the addition of TNF-α, and this may be due to its decreased dependency on classical NF-kB signaling that we previously showed [33,34].

Sufficient levels of TNF-α in the environment may render tumors particularly susceptible to IAP antagonism [35]. In our report, the HDAC inhibitor-induced TNF-α secretion corresponded to a synergistic anti-tumor effect with SMs. The synergy between birinapant and chemotherapeutic drugs, including docetaxel, gemcitabine, SN-38 (active metabolite of irinotecan), and 5-AC (5-azacytidine), has been previously reported [11,36]. Here, we describe a novel synergistic combination of SMs and HDAC inhibitors. Pretreatment with a neutralizing anti-TNF-α antibody [30] or Si-RNA of TNF-α [11] resulted in rescue from tumor cell death led by the birinapant and chemotherapeutic drugs. In the same way, we showed that depletion of TNF-α by RNA interference attenuated the synergistic effect between tolinapant and entinostat. This result suggests that HDAC inhibitor-induced TNF-α plays a critical role in the synergy.

A combination of tolinapant and entinostat was effective in vivo in several mouse models. Although we expected HDAC inhibitors to have some anti-tumor effect, neither panobinostat nor entinostat single agents showed anti-tumor activity in any of our models. A previous report indicated that the anti-tumor effect of entinostat single agent required adaptive immunity [15]. Even so, we did not observe improvement in survival of the C57B6 immune-competent mouse treated with entinostat + anti-PD1 antibody compared to entinostat alone or vehicle-treated mice. Both SMs and HDAC inhibitors have limited efficacy when used as single agents in ovarian cancer models, but the synergistic anti-tumor effect was confirmed in all mouse models. Furthermore, mouse survival improved with a triplet combination of tolinapant, entinostat, and anti-PD1 antibody, indicating the importance of immune modulation.

The comprehensive, unbiased matrix drug screen that identified this combination treatment is a powerful discovery technique to identify synergistic drug strategies that may improve outcomes for patients with ovarian cancer, and a clinical trial is currently under development. An understanding of the mechanisms underlying the synergy will allow for the discovery of novel therapeutic options for women with recurrent ovarian cancers that acquire apoptosis resistance due to IAP upregulation.

## 5. Conclusions

HDAC inhibitors mechanistically synergize with SMAC mimetics across a panel of ovarian cancer cell lines representing multiple histologic subtypes. The mechanism of synergy relies on stimulating autocrine TNF-α production that triggers extrinsic apoptosis when IAPs are depleted and SMAC is released from XIAP. The combination of a SMAC mimetic with an HDAC inhibitor is a novel and promising strategy for the treatment of cancers that resist apoptosis through the upregulation of IAP proteins and NF-kB signaling.

## Figures and Tables

**Figure 1 cancers-15-01315-f001:**
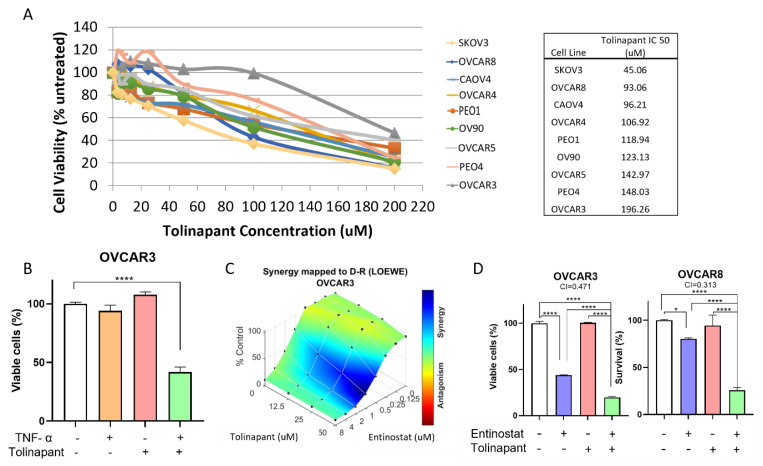
Tolinapant is synergistic in combination with an HDAC inhibitor in killing ovarian cancer cells. (**A**) Sensitivities to SMAC mimetic and tolinapant differ between ovarian cancer cell lines. Dose titration of tolinapant was carried out for 9 ovarian cancer cell lines (OVCAR3, OVCAR4, OVCAR5, OVCAR8, CAOV4, OV90, PEO1, PEO4, SKOV3) based on XTT cell viability assay. (**B**) Tolinapant has a synergistic anti-tumor effect with TNF-α. OVCAR3 cells were treated with external TNF-α (10 ng/mL) and/or tolinapant (25 μM) added simultaneously for 72 h. **** *p* < 0.0001 (**C**) Synergy between tolinapant and entinostat was confirmed in vitro using matrix titrations of both drugs in OVCAR3. Synergy analysis was performed with Combenefit. Blue surface indicates synergy. OVCAR3 cells were treated with entinostat (0–8 μM) and/or tolinapant (0–50 μM) added simultaneously for 72 h. (D) OVCAR3 and OVCAR8 cells were treated with tolinapant (25 μM) and/or entinostat (2 μM) for 72 h. * *p* < 0.05, **** *p* < 0.0001, combination index (CI) > 1 represents antagonism, CI < 1 represents synergism. CI is calculated with CalcuSyn.

**Figure 2 cancers-15-01315-f002:**
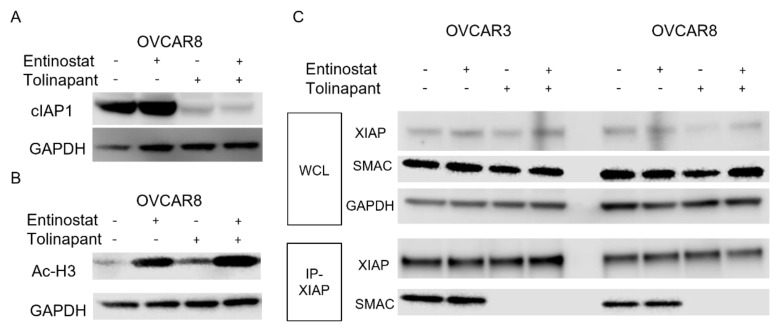
Tolinapant inhibits cIAP1 and XIAP, and entinostat acetylate Histone H3. (**A**,**B**) Changes in cIAP1 and acetylated histone H3 were measured with Western blot in OVCAR8 after 24 h treatment with either single-agent or combination tolinapant 25 μM and entinostat 2 μM. cIAP1 was degradated by tolinapant. Histone H3 was acetylated with entinostat treatment. (**C**) Changes in protein–protein interaction between SMAC and XIAP Western blot after co-immune precipitation using anti XIAP antibody in OVCAR3 and OVCAR8 after 24 h treatment with either single-agent or combination tolinapant 25 μM and entinostat 2 μM. Tolinapant did not change the protein level of XIAP and SMAC, but conjugation between SMAC and XIAP was completely inhibited.

**Figure 3 cancers-15-01315-f003:**
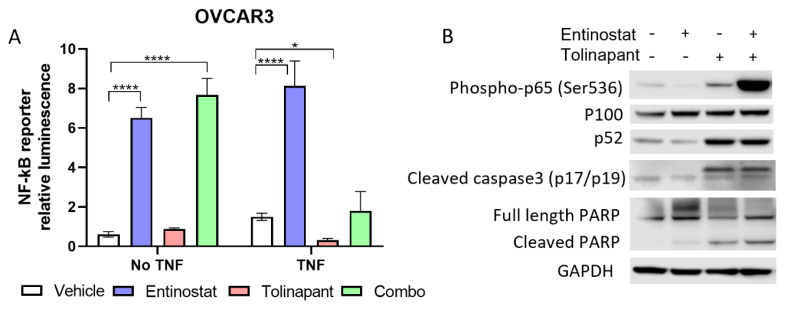
HDAC inhibitors and SMAC mimetics synergistically increase cell death through NF-kB activation. (**A**) NF-kB signaling was assessed in OVCAR8 cells after establishing a reporter cell line stably transduced with a lentiviral vector containing luciferase under control of the NF-kB consensus response element. Reporter cells were treated with single-agent (tolinapant 25 μM, entinostat 2 μM) or combination therapy for 24 h, both with and without exogenous TNF-α (10 ng/mL). * *p* < 0.05, **** *p* < 0.0001 (**B**) Changes in NF-kB and apoptosis proteins were measured with Western blot in OVCAR8 after 24 h treatment with either single-agent or combination tolinapant 25 μM and entinostat 2 μM.

**Figure 4 cancers-15-01315-f004:**
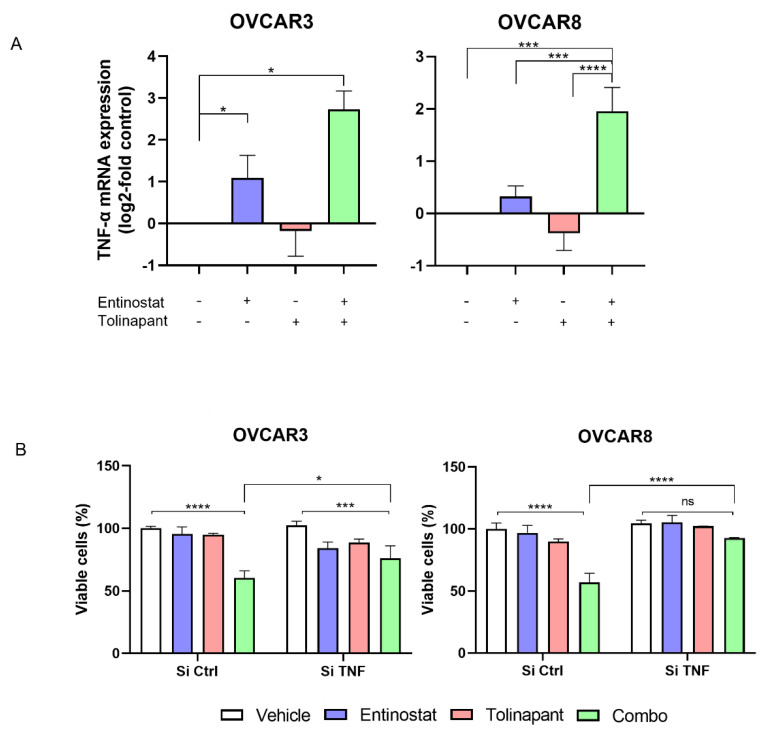
HDAC inhibitor Induced TNF-α secretion in tumor cells is critical for synergy with SMs. (**A**) TNF-α mRNA expression after tolinapant (25 μM) and/or entinostat (2 μM) 24 h treatments. (**B**) TNF-α knockdown using si-RNA canceled the combination effect of tolinapant (25 μM) and/or entinostat (2 μM) 72 h treatments. * *p* < 0.05, *** *p* < 0.001, **** *p* < 0.0001, ns: not significant.

**Figure 5 cancers-15-01315-f005:**
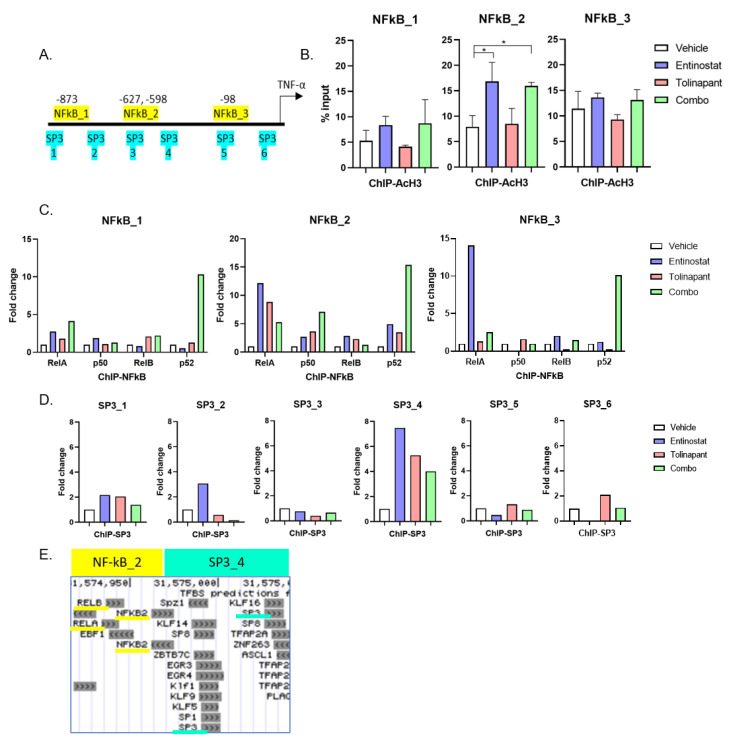
Entinostat enhances the production of TNF-α through acetylation of TNF-α promoter region. (**A**) There are three NFkB binding sites and 6 SP3 binding sites on the TNF-α promoter region. (**B**) Chromatin-IP (ChIP) qPCR assay using anti-acetylated histone H3 (AcH3) antibody. All isotype controls (IgG) were under 0.01% of input. DNA samples we corrected from OVCAR8 cells after tolinapant (25 μM) alone or in combination with entinostat (2 μM) 18 h treatment. (* *p* < 0.05). (**C**) ChIP qPCR assay using anti-NFkB (RelA, p50, RelB, p52) antibodies. DNA samples we corrected from OVCAR8 cells after tolinapant (25 μM) alone or in combination with entinostat (2 μM) 18 h treatment. (**D**) ChIP qPCR assay using anti-SP3 antibody. DNA samples we corrected from OVCAR8 cells after tolinapant (25 μM) alone or in combination with entinostat (2 μM), 18 h treatment. (**E**) Genomic locations of transcription factor binding sites in the promoter region for the TNF-α gene.

**Figure 6 cancers-15-01315-f006:**
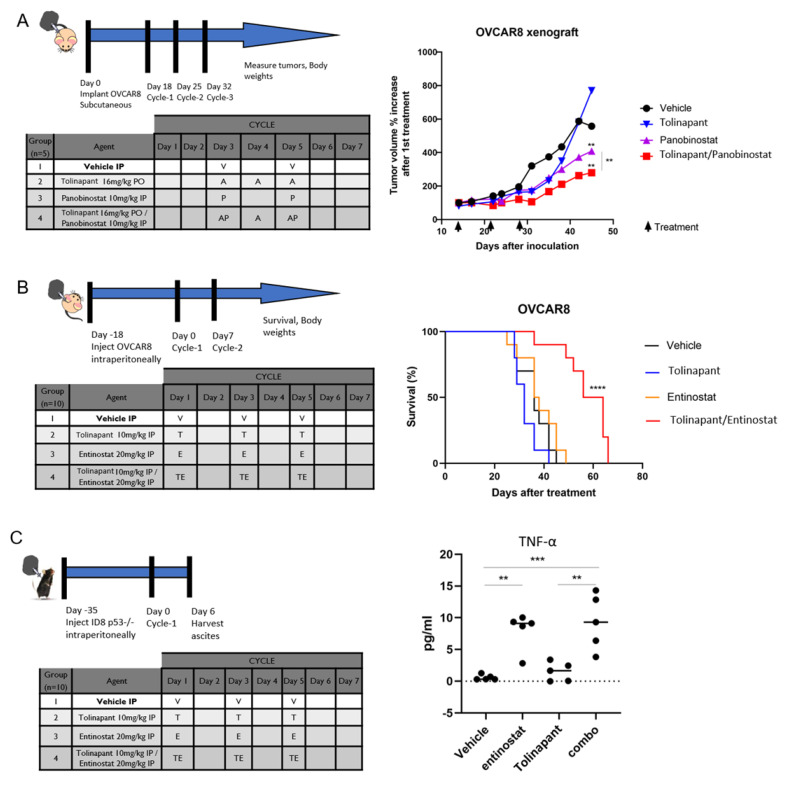
HDAC inhibitors are synergistic with tolinapant in vivo. (**A**) Athymic nude mice were inoculated subcutaneously with OVCAR8 ovarian cancer cells. Mice were randomized into treatment groups after tumors achieved an average volume of 50–100 mm^3^. Average total tumor volumes over time were measured by calipers following 3 weeks of vehicle, single agent tolinapant (16 mg/kg, oral), panobinostat (10 mg/kg, intraperitoneal, IP), or combination, as shown in the inset table. (**B**) OVCAR8 cells were inoculated IP into athymic nude mice. Survival was monitored after receiving 3 weekly IP treatments of single or combined tolinapant 10 mg/kg and entinostat 20 mg/kg. Log-rank test was performed to determine statistically significant differences in survival. **** *p* < 0.0001 (**C**) C57B6 mice were inoculated with ID8-p53KO syngeneic cell line. After 18 days, mice were treated in groups of 10 with vehicle, tolinapant, entinostat, or combination. Mice were euthanized after one week of treatment, and tissues were harvested for correlative studies. TNF-α was measured in mouse ascites with ELISA. ** *p* < 0.01, *** *p* < 0.001.

**Figure 7 cancers-15-01315-f007:**
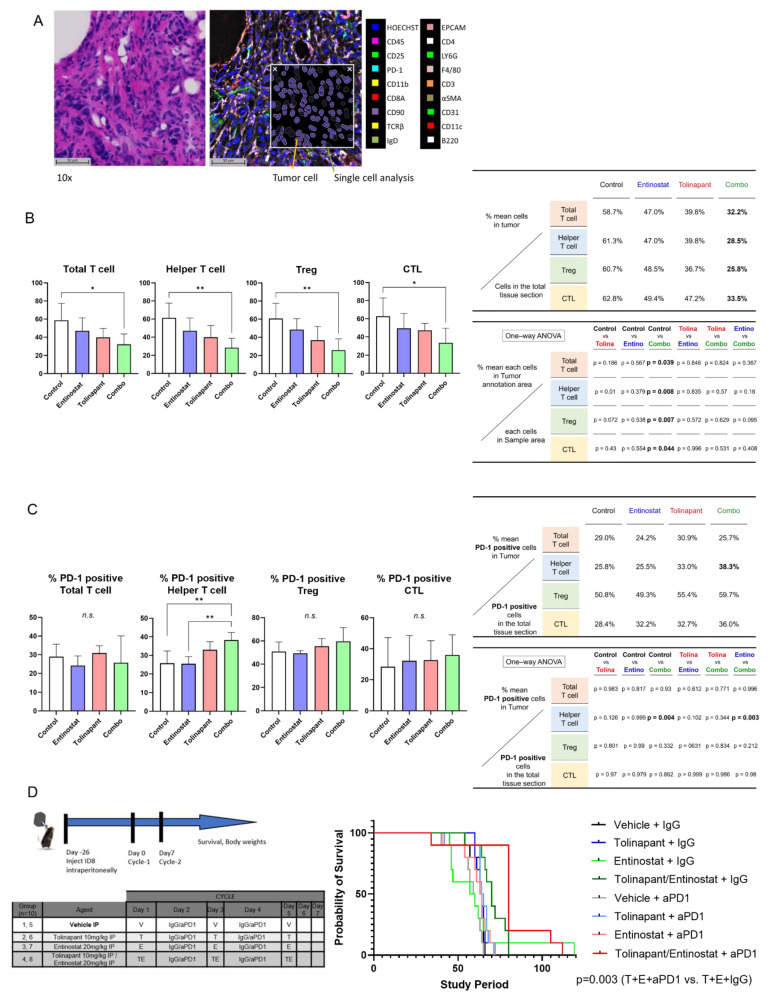
Characterization of immune infiltrates shows changes in T cell subsets with combination treatment. (**A**) example of tissue section stained with H&E; staining of tissue with CODEX technology to detect immune cell differentiation markers; insert shows single cell analysis using HALO 2D digital pathology analysis software and detection of tumor cells. (**B**) Changes in T cell subsets. (**C**) Changes in PD1 expression on T cell subsets. (**D**) Addition of anti-PD1 antibody improves survival in immune-competent mice. * *p* < 0.05, ** *p* < 0.01.

**Figure 8 cancers-15-01315-f008:**
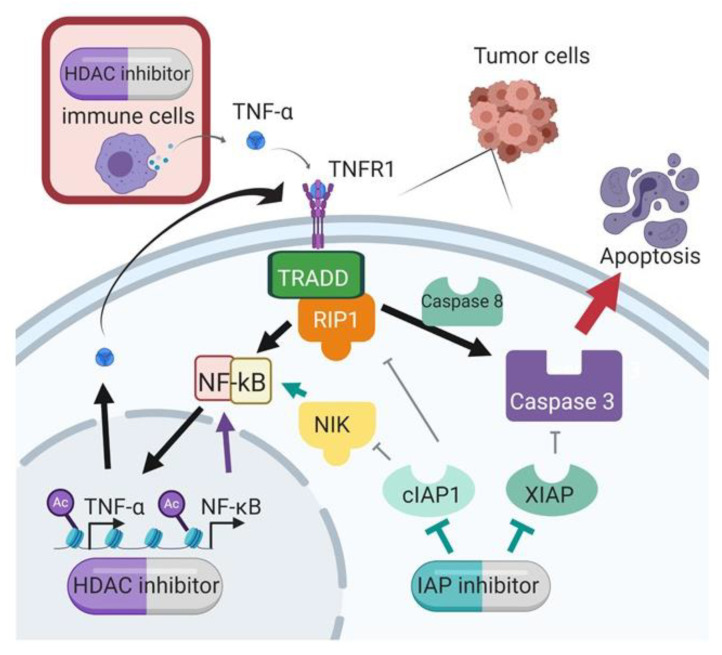
Proposed mechanism of synergy between SMAC mimetics and HDAC inhibitors.

## Data Availability

Not applicable.

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
