# Peer review of "SMAC Mimetics Synergistically Cooperate with HDAC Inhibitors Enhancing TNF-α Autocrine Signaling"

_cancers, 2023, doi:10.3390/cancers15041315_

Round 1

Reviewer 1 Report

The work done by Shibuya et al is a commendable piece of research that aims at addressing the synergistic effect of SMAC mimetics and HDAC inhibitors in Ovarian cancer.  To start with the positive aspects of the manuscript - good hypothesis, use of advanced techniques and asking the right questions. The authors need to be commended for that.

Even then there are some major issues that need to be addressed before the manuscript can be accepted for publication in Cancers. The first and foremost issue that I have with this manuscript is the lack of focus. The data is all over the place. The use of different cell lines and compounds (even within the same figure) takes away the beauty of the work. Though the authors try to justify this in the discussion  (lines 581 -583) as not individual agent/cell line effect - it lacks focus.

The major issues with the manuscript:

1. Please keep the drug combination and cell lines consistent through out the study. And to show that the effect is not agent or cell line specific - include those data as Suppl.info. And there is also a lack of explanation/rationale when the drug pairs or cell lines are changed - even within a single figure!

2.  Regarding the drug addition, what is the rationale for adding it together? If it is expected that SM exerts the effect first, and HDACi adds to it ?

Similarly for the exogenous TNFa addition expts, was there a pretreatment with TNFa? Please include in text.

3. Check lines 360 -363 - the conclusion drawn from Fig3.C is wrong. "entinostat and tolinapant activated NF-kB function in the 360 absence of TNF-α...." The effect seems to be dependent more on the small molecule inhibitor used. This is not discussed. Please include. 

4. Lines 387 -389 is wrong - "This suports the hypothesis that the sensitivity of TOV21G and SKOV3 to tolinapant single treat-388 ment is dependent on TNF-α autocrine secretion" - mainly because the viability is variable in Fig.1. In Fig1 D - SKOV3 when treated with 25uM the cell viablity varies from nearly 40% to 75%. This is not regarding the combination treatments, but the  Tolinapant single agent treatment. Please rectify this mistake. 

5. As stated above, OVCAR3 also exhibits such variation in cell viability in Fig.1 B and D - where in response to Tolinapant the viability is 100% in 1B, but reduces to 75% in 1D (single agent treatment).

6. Lines 479 -480 - conclusion is wrong. "mice treated with the combination of 479 tolinapant, entinostat, and anti-PD1 antibody had the longest median survival" - Entinostat treatment alone has the highest survival as per the figure. Please change the color format of the graph as Entinostat +/- a-PD1 has the same color. Difficult to comprehend.

7.  After Fig 1 where different cell lines are tested, there is no rationale or explanation for doing the effect on cIAP/XIAP in OVCAR cells. This trend is all through the manuscript - where there is no explanation for jumping from one cell line/drug to another. 

8. The effect on cIAP and the interaction with XIAP - does it change in response to exogenous TNFa addition? 

9. Where the authors surprised to see the difference in response of OVCAR3 and 8 - in Fig 3 - in terms of Caspase activation in the absence of TNFa? No explanation was included. Please address this difference of observation.

10. Also Fig. D - why TOV21-G all of a sudden?? And the western blot shows Entinostat and Tolinapant on OVCAR8 for which there is no data shown to check in the NFkB activation in reporter cell lines. Please keep consisitent.

11. What is the percentage of knockdown seen after siRNA treatment in Fig.4? No data provided on that. Also no explanation/observation has been given with regard to the difference in the mRNA and protein levels of TNFa. Also as seen throughout the manuscript, the drug combinations keep on changing. Please keep it consistent.

12. Does the cell viability in Fig. 4C reflect the data in 4A and B?

13. Fig 6 - does not provide any information on whether the mice developed any tumor? Ascitic fluid accumulation? Asc.fluid volume? Abdominal circumference?

Minor issues:

1. Methodology 2.5 and 2.6 are duplicates

2. Lines 434-436 - Check for grammatical errors

3. Suppl. info missing, available only from Suppl. fig 8.  Please provide the rest of the figures. 

4. Check fig 1C? The matrix provided for Vorinostat (?) and Tolinapant? Is it a continuation of 1C? Include in Fig.legend.

 5. For the column graphs, please provide some kind of distinguishable feature like color or pattern. Will be easy to follow the data then.

6. Fig 5 - 5E missing in Fig.legend. Keep consistency - Tolinapant instead of ASTX.

Reviewer 2 Report

This manuscript entitled SMAC mimetics synergistically cooperate with HDAC inhibitors enhancing TNF-α autocrine signaling by Shibuya et al. deals with exploring the synergistic mechanism between SMs and HDAC inhibitors as a promising strategy for cancer treatment. A detailed and careful review of the manuscript leads me to make the following comments:

1. There are several grammatical errors in the manuscript that the authors need to take care of before submission to any journal. As a result, there is a need to improve the English throughout the manuscript. Some sentences are not clear. Further, standard journal guidelines and formatting should be followed before submission. Specifically, there are similarities between the current text and prior studies. Thus, the whole text should be rewritten.

2. Section 2.5 has been duplicated. So, delete it and rearrange the numbering.

3. Why were OVCAR3 and OVCAR8 cell lines selected for the rest of the study?

4. Regarding the concentration used, the authors should explain the reason for selecting each concentration. It is not clear on what basis the concentrations are selected.

5. To allow for clinical translation, the doses used should be comparable to those commonly used in clinical trials. Are your selected doses recommended for preclinical studies? See the reference:

D. R. Liston and M. Davis, Clinically Relevant Concentrations of Anticancer Drugs: A Guide for Nonclinical Studies Clinical Cancer Research 2017, 23(14), 3489.3498. http://doi.org/10.1158/1078-0432.ccr-16-3083

6. The range of the graphs must be adapted so that comparability can be done.

7. The conclusion is so short. Please rewrite this section and mention the main points there.

The present format of the manuscript requires revisions before being reviewed further and extensively. Therefore, I do not recommend this document for publication in this journal in its present form.

Round 2

Reviewer 1 Report

The line numbers provided in the revised manuscript do not match with those given in the response letter. 

Reviewer 2 Report

The manuscript is accepted as the authors have fulfilled all the queries.